# Imipramine, an Acid Sphingomyelinase Inhibitor, Promotes Newborn Neuron Survival in the Hippocampus After Seizure

**DOI:** 10.3390/cells14040281

**Published:** 2025-02-14

**Authors:** Song Hee Lee, Hyun Wook Yang, Beom Seok Kang, Min Kyu Park, Dong Yeon Kim, Hong Ki Song, Hui Chul Choi, Minwoo Lee, Bo Young Choi, Dae-Soon Son, Sang Won Suh

**Affiliations:** 1Department of Physiology, College of Medicine, Hallym University, Chuncheon 24252, Republic of Korea; 2thgml@naver.com (S.H.L.); akqjqtj5@hallym.ac.kr (H.W.Y.); ttiger1993@gmail.com (B.S.K.); bagmingyu50@gmail.com (M.K.P.); roy8596@naver.com (D.Y.K.); 2Department of Neurology, Kangdong Sacred Heart Hospital, Seoul 05355, Republic of Korea; hksong0@hanmail.net; 3Department of Neurology, Hallym University Sacred Heart Hospital, Chuncheon 24253, Republic of Korea; dohchi@naver.com; 4Department of Neurology, Hallym University Sacred Heart Hospital, Anyang 14068, Republic of Korea; minwoo.lee.md@gmail.com; 5Department of Physical Education, Hallym University, Chuncheon 24253, Republic of Korea; bychoi@hallym.ac.kr; 6Division of Data Science, Data Science Convergence Research Center, Hallym University, Chuncheon 24253, Republic of Korea; biostat@hallym.ac.kr; 7Hallym Institute of Epilepsy Research, Chuncheon 24253, Republic of Korea

**Keywords:** epilepsy, imipramine, ceramide, acid sphingomyelinase, neuron death, neurogenesis, cognitive function

## Abstract

Epilepsy, a chronic neurological disorder, is triggered by various insults, including traumatic brain injury and stroke. Acid sphingomyelinase (ASMase), an enzyme that hydrolyzes sphingomyelin into ceramides, is implicated in oxidative stress, neuroinflammation, and neuronal apoptosis. Ceramides, which have pro-apoptotic properties, contribute to oxidative damage and lysosomal dysfunction, exacerbating neuronal injury. This study investigates the role of ASMase in epilepsy, hypothesizing that seizure activity upregulates ASMase, increasing ceramide levels, DNA damage, and neuronal apoptosis. We employed a pilocarpine-induced rat seizure model and examined the effects of imipramine, an ASMase inhibitor, administered intraperitoneally (10 mg/kg) for four weeks post-seizure induction. Histological and cognitive analyses showed that while imipramine did not prevent early neuronal death within the first week, it significantly reduced markers of neuronal apoptosis by four weeks. Imipramine also promoted hippocampal neurogenesis and preserved cognitive function, which is often impaired following seizures. These findings suggest that ASMase inhibition could mitigate neuronal apoptosis and improve cognitive recovery after seizures. Imipramine may serve as a promising therapeutic strategy for epilepsy-associated neuronal damage and cognitive deficits. Further studies should delineate the molecular mechanisms of ASMase inhibition and evaluate its long-term efficacy in addressing epilepsy-related neurodegeneration and functional impairments.

## 1. Introduction

Epilepsy is a chronic neurological disorder characterized by recurrent, spontaneous seizures resulting from abnormal neuronal activity in the brain’s cortex [1,2,3,4]. While the manifestations of epilepsy vary widely, causes such as brain injury, stroke, tumors, and infections have been associated with its development [5,6]. However, despite extensive research, the precise mechanisms underlying epilepsy remain poorly understood, making it difficult to address the full spectrum of the disorder’s effects [2,7,8].

One significant consequence of chronic seizures is their impact on neurogenesis, particularly in the hippocampus, a critical region involved in memory and learning [9,10,11,12]. In the early stages of chronic seizures, neurogenesis may initially increase as the brain attempts to compensate for neuronal damage [11]. However, prolonged seizure activity disrupts this process, leading to diminished neuronal integration, widespread neuronal death, and alterations in the neurogenic environment [10]. These changes can severely affect cognitive function, with deficits frequently observed in individuals with chronic epilepsy. The interaction between seizure activity and neurogenesis is complex, influenced by factors such as seizure frequency, severity, and individual variability.

In the adult brain, neurogenesis occurs in the hippocampal dentate gyrus, where neural progenitor cells give rise to new neurons [13,14,15]. These newly formed neurons integrate into existing brain circuits and play a vital role in maintaining learning and memory [16,17,18]. However, factors such as oxidative stress from aging, traumatic brain injury, or epilepsy can disrupt neurogenesis [19,20,21,22]. This emphasizes the need for protective mechanisms to preserve the function of neural progenitors and to support the ongoing process of neurogenesis in the face of chronic neurological stressors.

Sphingomyelin, a lipid component of the plasma membrane, is metabolized into ceramide by the enzyme sphingomyelinase [23,24]. Under normal physiological conditions, ceramide plays an important role in regulating various cellular functions, including cell proliferation, differentiation, and apoptosis [25,26]. However, in disease states, the overactivation of sphingomyelinase can lead to excessive ceramide production [27,28]. Elevated ceramide levels have been linked to increased apoptosis, inflammation, and oxidative stress, all of which contribute to neuronal death [29]. Importantly, this pathological increase in ceramide has been associated with impaired neurogenesis in the hippocampus, further complicating the brain’s ability to recover from seizures [30].

Tricyclic antidepressants (TCAs), which are commonly used for treating depression, are also known to inhibit acidic sphingomyelinase activity [31,32,33]. Imipramine, a TCA, has shown neuroprotective effects through this mechanism, reducing neuron death and cognitive decline in various neuropathological conditions, including traumatic brain injury [34]. Studies have demonstrated that imipramine can preserve neurogenesis by promoting the maturation of neural progenitor cells into functional neurons and astrocytes, thus supporting hippocampal integrity [35,36].

This study aims to explore the potential of imipramine as an inhibitor of acidic sphingomyelinase in reducing neuronal damage and mitigating cognitive dysfunction in a rat model of pilocarpine-induced seizures. By investigating the role of imipramine in enhancing neuronal survival and neurogenesis, this research seeks to identify new therapeutic approaches for treating epilepsy-related cognitive impairments.

## 2. Materials and Methods

### 2.1. Ethics Statement and Experimental Animals

This study was conducted with the approval of the Animal Research Committee at Hallym University’s College of Medicine, following established ethical guidelines for animal experimentation (Protocol No.: Hallym R1(2021-9)). The research utilized adult male Sprague–Dawley rats obtained from DBL Co. in Chungcheongbuk-do, Republic of Korea. The rats, approximately 8 weeks old and weighing between 300 and 350 g, were housed under carefully controlled environmental conditions. These conditions included a 12 h light/dark cycle, with temperatures maintained at 20 ± 2 °C and humidity levels kept at 55% ± 5%. These measures ensured that the rats were provided with a stable, consistent, and ethically compliant environment throughout the study period.

### 2.2. Pilocarpine-Induced Seizure Model

To induce seizures, rats were administered an intraperitoneal injection of pilocarpine (25 mg/kg) in accordance with a standard protocol [37,38,39,40,41]. Muscarinic receptor activity was enhanced by a prior intraperitoneal injection of lithium chloride (LiCl, 127 mg/kg). To counter excessive salivation, a common side effect of pilocarpine, scopolamine (2 mg/kg) was administered 30 min before the pilocarpine injection. The severity of status epilepticus (SE) was evaluated using the Racine stage classification, which categorizes seizure severity into five stages based on observed behaviors such as facial movements, rearing, and falling. In this model, the rats typically exhibited phase 4 or higher seizures, characterized by rearing, indicating successful induction of seizures [37,42].

To manage the seizures, diazepam (10 mg/kg), a standard antiepileptic drug, was injected intraperitoneally one hour after SE onset. This approach allowed for controlled seizure induction and effective evaluation of SE severity, followed by the administration of diazepam to manage and control seizure activity.

### 2.3. Brain Sample Preparation

Rats subjected to seizures were euthanized at two key time points: 24 h and 7 days post-seizure. Euthanasia was performed using urethane anesthesia (1.5 g/kg, intraperitoneally), followed by perfusion with 0.9% saline. After perfusion, the brains were fixed in 4% paraformaldehyde to ensure tissue preservation. Following fixation, the brains were kept in 4% paraformaldehyde for one hour before being transferred to a 30% sucrose solution until they were fully saturated. For histological analysis, the brains were sectioned into 30 μm-thick slices using a CM1850 cryostat microtome (Leica, Wetzlar, Germany), and this process was carried out two days after fixation.

### 2.4. Detection of Live Neurons

To evaluate the neuroprotective effects of imipramine following seizures, NeuN immunohistochemistry was conducted on brain sections at 7 days and 4 weeks post-seizure induction. The brain tissues were sliced into 30 µm sections for this analysis. After rinsing with PBS buffer, the sections were incubated with mouse anti-NeuN primary antibodies (1:500, Millipore, Billerica, MA, USA) in PBS containing Triton X-100. Secondary antibodies (anti-mouse IgG, 1:250; Vector, Burlingame, CA, USA) were then applied as part of the immunohistochemistry process. For visualization, the sections were treated with ABC solution (Vector, Burlingame, CA, USA) and stained using DAB (3,3′ diaminobenzidine, Sigma-Aldrich, St. Louis, MO, USA) for 1 min and 30 s. The stained sections were mounted on slides with DPX mounting medium and examined under an Olympus IX70 inverted microscope (Olympus Co., Tokyo, Japan) to identify NeuN-positive cells. Neuronal identification and differentiation from non-specific background signals were analyzed using Image J software(Version 1.53t).

### 2.5. Detection of Oxidative Stress in the Hippocampal Region

For the 4-hydroxyl-2-nonenal (4HNE) immunohistochemistry assay, brain tissue samples were initially cleansed through three 10 min washes in 0.01 M PBS. To remove any residual blood from the tissues, a pretreatment procedure was applied, consisting of sequential applications of distilled water, 90% methanol, and 30% hydrogen peroxide, each for 15 min. Following pretreatment, the samples were washed three additional times in 0.1 M PBS, with each wash lasting 10 min.

Next, the tissues were incubated overnight at 4 °C with a primary antibody solution targeting 4HNE. This solution was prepared using mouse anti-4HNE serum diluted 1:500 in PBS containing 0.3% Triton X-100 (Alpha Diagnostic Intl. Inc., San Antonio, TX, USA). After the primary antibody incubation, the samples were rinsed three times in 0.01 M PBS, with each rinse lasting 10 min. Subsequently, the tissues were incubated with a secondary antibody solution from Invitrogen (Grand Island, NY, USA), containing donkey anti-mouse IgG conjugated to Alexa-Fluor-594 at a 1:250 dilution. This incubation was carried out at room temperature for two hours.

Following the secondary antibody incubation, the samples were washed three additional times in 0.01 M PBS for 10 min each. The tissues were then mounted on slides and cover-slipped using DPX mounting media (Sigma-Aldrich, St. Louis, MO, USA). Fluorescence observation was performed using an Olympus fluorescence microscope (Tokyo, Japan).

Quantification of 4HNE staining was performed by analyzing fluorescence intensity using ImageJ software (National Institutes of Health, Bethesda, MD, USA). The analysis focused on predefined regions of interest in the tissue sections. The mean fluorescence intensity of each region was calculated and averaged to determine the overall fluorescence intensity for each experimental group.

### 2.6. Evaluation of ASMase Activation and Ceramide Formation

To examine the effects of imipramine on ASMase and ceramide levels 4 weeks after seizure induction, immunofluorescence labeling was conducted on brain tissue sections. Initially, the sections were washed three times with 0.01% phosphate-buffered saline (PBS) for 10 min each to prepare them for staining. The primary antibodies used were rabbit anti-ASMase (diluted 1:100, sourced from Invitrogen, Grand Island, NY, USA) and mouse anti-ceramide (diluted 1:10, from Enzo Life Science, Enzo Biochem, Inc., Farmingdale, NY, USA). These antibodies were applied for targeted immunofluorescence staining. Following primary antibody application, the brain sections were mounted on slides using DPX mounting medium.

The expression of ceramide and ASMase in the brain tissue was then analyzed using a fluorescence microscope. This allowed for the assessment of imipramine’s effects on ASMase activity and ceramide levels in the post-seizure condition.

### 2.7. Western Blot

Western blotting was performed to assess the protein levels of Bax, Bcl2, Akt, and p-Akt in the hippocampus for both the vehicle and imipramine-treated groups. Following saline cerebral perfusion, hippocampal tissues were dissected and homogenized in RIPA buffer. The homogenate was incubated on ice for 30 min, followed by centrifugation at 14,000 rpm for 20 min at 4 °C. The supernatant was collected and stored at −80 °C for further analysis. Protein concentrations were determined using the Bradford protein assay.

For the Western blot, proteins from the supernatant were diluted with SDS sample buffer, separated on 8–12% SDS-polyacrylamide gels, and transferred onto PVDF membranes. To prevent non-specific binding, the membranes were blocked with 5% skim milk for 1–2 h. Primary antibodies were then incubated overnight at 4 °C, including monoclonal mouse anti-Bax (1:1000, sc-7480), monoclonal mouse anti-Bcl2 (1:1000, sc-7382), monoclonal rabbit anti-Akt (1:1000, Cell Signaling #9272, Danvers, MA, USA), and monoclonal rabbit anti-p-Akt (1:2000, Cell Signaling #4060). After primary antibody incubation, the membranes were washed three times with TBST for 10 min each and incubated with secondary antibodies (anti-rabbit IgG and anti-mouse IgG, 1:5000 dilution; Ab Frontier, Seoul, Korea) for 1 h at room temperature.

Protein bands were visualized using a chemiluminescence bioimaging system (Amersham Imager 680, Marlborough, MA, USA).

### 2.8. Detection of BrdU Labeling

To assess the long-term survival and differentiation of newly generated cells influenced by imipramine, the experimental group was injected with BrdU at 12 h intervals for four consecutive days. To detect BrdU-labeled cells in tissue sections, an initial treatment with 0.3% hydrogen peroxide in methanol was performed for 15 min to enhance cell visualization through immunoperoxidase staining. Following this, the sections underwent DNA denaturation in 2 N HCl for 90 min to prepare them for BrdU immunostaining. After the HCl treatment, the sections were neutralized with 0.1 M sodium borate buffer using two 10 min rinses. Each step was followed by three 10 min washes with phosphate-buffered saline (PBS). The primary antibody, monoclonal BrdU (diluted 1:150, Roche Co., Basel, Switzerland), was applied at room temperature for one hour in a solution containing 1% normal chicken serum and Triton X-100. After binding of the primary antibody, the sections were incubated for two hours with a secondary anti-mouse IgG antibody (diluted 1:250). Visualization was achieved by a two-hour incubation with a biotin-peroxidase complex (ABC, Vector Lab., Burlingame, CA, USA), followed by staining with 3,3′-diaminobenzidine (DAB, Sigma-Aldrich Co., St. Louis, MO, USA).

### 2.9. Behavior Tests

The Barnes maze test was used to evaluate the recovery of spatial cognitive function in rats following seizure induction, specifically assessing the effects of amlodipine treatment. Four weeks after epilepsy induction, the rats were divided into two groups: one group received immediate imipramine treatment following seizure onset, while the other group experienced a delayed initiation of therapy. The Barnes maze consisted of a circular platform with multiple holes, only one of which led to an escape chamber. During the test, rats were placed in a black cylinder for acclimatization and then subjected to a disorienting task. The time taken to locate the escape hole was recorded. Each trial lasted a maximum of 2 min, and failure was defined as the inability to find the escape hole or falling off the platform within this time frame.

The behavioral testing was conducted over five consecutive days. Data collection and statistical analysis were performed using IBM SPSS Statistics version 29.0.2.0. Following the testing period, the rats were euthanized, and their tissues were collected and processed according to previously described methods.

### 2.10. Data Analysis

Data analysis in this study was performed using Image J software (National Institute of Health, Bethesda, MD, USA), with results expressed as mean ± standard error of the mean (SEM). The Mann–Whitney U test was used to compare imipramine treatment and vehicle control groups. This non-parametric test does not assume normality of the data distribution, making it appropriate for our dataset, in which normality could not be reliably confirmed due to small sample sizes. A *p*-value of less than 0.05 was considered statistically significant. Additionally, IBM SPSS statistical software was employed for more comprehensive data analysis.

## 3. Results

### 3.1. Impact of Imipramine on Hippocampal Neuron Survival After Seizure

We assessed the neuroprotective effects of imipramine in a pilocarpine-induced seizure model by using NeuN labeling to quantify surviving neurons in the hippocampus 1 week post-seizure. First, in this experiment, imipramine was injected intraperitoneally (i.p.) at a concentration of 10 mg/kg after the seizures, as in our previous studies. [34,43]. Significant neuronal loss was observed in hippocampal regions, including the hilus, CA3, CA1, and subiculum, following the seizure (Figure 1B). Immunohistochemistry with anti-NeuN antibodies was used to compare neuron survival between the seizure vehicle group and the seizure imipramine group. At the one-week point, no significant differences in NeuN-positive neurons were detected between these groups (Figure 1C), suggesting that imipramine does not exert neuroprotective effects immediately after seizure induction. However, after a continuous four-week course of imipramine treatment, significant differences in neuron survival emerged. NeuN-positive neurons were notably increased in the imipramine-treated group compared to the seizure vehicle group (Figure 1D,E). Specifically, the imipramine group exhibited a 64.9% increase in NeuN-positive neurons in the CA1 region, a 37.4% increase in CA3, a 36% increase in the dentate gyrus (DG), and a 34% increase in the subiculum (Sub). These findings suggest that prolonged imipramine treatment promotes neuron survival following a pilocarpine-induced seizure. These results indicate that a four-week imipramine regimen may significantly enhance neuronal survival after seizures, highlighting imipramine’s potential as a therapeutic agent for mitigating seizure-induced neuronal damage and promoting recovery following status epilepticus.

### 3.2. Imipramine Decreased Oxidative Stress After Pilocarpine-Induced Seizure

We examined the effectiveness of imipramine in reducing oxidative stress in a pilocarpine-induced seizure model by using 4-hydroxy-2-nonenal (4HNE) immunohistochemical staining to assess oxidative damage in the hippocampus at 1 and 4 weeks after imipramine administration. Minimal 4HNE staining was detected in both the saline and imipramine-treated sham groups. In contrast, the pilocarpine-induced seizure group showed a significant increase in 4HNE fluorescence intensity in the hippocampus compared to the sham group, particularly in the CA1, CA3, hilus, and subiculum regions, as shown in Figure 2B.

At the 1-week mark, there was no significant difference in 4HNE fluorescence intensity between the seizure imipramine and seizure vehicle groups (Figure 2A,B). However, at the 4-week evaluation, the seizure vehicle group exhibited a marked increase in 4HNE intensity in the hippocampus, as illustrated in Figure 2C. In contrast, the seizure imipramine-treated group demonstrated a significant reduction in 4HNE intensity compared to the vehicle-treated group, as shown in Figure 2D. Imipramine treatment led to a notable decrease in 4HNE intensity of 42.8% in CA1, 42.3% in CA3, and 39.9% in the hilus. While a reduction in 4HNE intensity was observed in the subiculum, it did not reach statistical significance.

### 3.3. Effect of Imipramine on Acid Sphingomyelinase Activity and Ceramide Production Post-Seizure

To investigate the effects of imipramine on acid sphingomyelinase (ASMase) activity and ceramide production post-seizure, immunofluorescence staining was performed four weeks after seizure induction.

In the sham groups, there were no significant differences in ASMase levels between vehicle- and imipramine-treated animals. However, in the seizure groups, vehicle-treated animals exhibited a significant increase in ASMase expression compared to the sham groups. Imipramine treatment effectively attenuated this seizure-induced upregulation of ASMase, as evidenced by reduced ASMase fluorescence intensity in the hippocampal CA1 region (Figure 3A). Quantitative analysis revealed ASMase levels as follows: 10.6 ± 0.4 in the sham vehicle group, 11.31 ± 1.1 in the sham imipramine group, 18.2 ± 1.1 in the seizure vehicle group, and 13.2 ± 0.4 in the seizure imipramine group, representing a 27.6% reduction in ASMase levels in the seizure imipramine group compared to the seizure vehicle group (Figure 3A,B).

Ceramide levels, assessed using anti-ceramide antibody staining, showed a similar trend. No significant differences were observed between the sham groups. However, in the seizure groups, increased ASMase activity correlated with elevated ceramide production, particularly in the seizure vehicle group. Imipramine treatment reduced ceramide levels in the CA1 region, indicating effective inhibition of ASMase activity. Ceramide levels were quantified as follows: sham vehicle, 11.6 ± 2.9; sham imipramine, 11.7 ± 0.3; seizure vehicle, 31.6 ± 4.6; seizure imipramine, 14.5 ± 1.0. This reflects a 53.8% reduction in ceramide levels in the seizure imipramine group compared to the seizure vehicle group (Figure 3C,D). Although this reduction in ceramide production approached statistical significance (*p* = 0.066), it is likely that the sample size in this study was insufficient to achieve significance.

These findings suggest that imipramine effectively reduces seizure-induced ASMase activity and ceramide overproduction in the hippocampus. These results highlight imipramine’s potential to modulate ASMase and ceramide levels, providing insights into its therapeutic effects post-seizure.

### 3.4. Effects of Imipramine on Apoptosis and Proliferation Markers Post-Seizure

Seizures often lead to increased ceramide synthesis, primarily through ASMase overexpression, which triggers apoptosis. The Bax/Bcl-2 ratio is a key determinant of cell fate, with Bax promoting cell death by permeabilizing the mitochondrial outer membrane and Bcl-2 inhibiting this process [44,45,46]. In contrast, phosphorylated Akt (p-Akt) plays a crucial role in promoting cell survival and proliferation [47,48,49,50].

In our study, no significant differences were observed in the Bax/Bcl-2 ratio or p-Akt levels between the sham vehicle and sham imipramine groups. However, in the post-seizure groups, the seizure imipramine group showed a notable reduction in the Bax/Bcl-2 ratio, indicating decreased apoptosis (a 63.8% decrease in the Bax/Bcl-2 ratio, Figure 4A,B). Additionally, p-Akt levels, which were lower in the seizure vehicle group, increased significantly in the seizure imipramine group, with a 37.7% rise in p-Akt levels (Figure 4). The detailed ratios are as follows: Bax/Bcl2—sham vehicle = 1.0 ± 0.1, sham imipramine = 1.15 ± 0.1, seizure vehicle = 3.59 ± 0.6, seizure imipramine = 1.3 ± 0.3; p-Akt—sham vehicle = 1.0 ± 0.2, sham imipramine = 1.17 ± 0.2, seizure vehicle = 0.54 ± 0.1, seizure imipramine = 0.87 ± 0.

These findings suggest that imipramine treatment post-seizure can effectively reduce apoptosis by lowering the Bax/Bcl-2 ratio and promoting cell survival and proliferation, as reflected by the increased p-Akt levels. This highlights imipramine’s potential role in enhancing the survival of newly generated cells following seizures.

### 3.5. Influence of Imipramine on Progenitor Cell Survival Post-Seizure

We assessed the effects of 1-week and 4-week imipramine treatments on the generation and survival of progenitor cells in the hippocampus following seizures. Using 5-bromo-2′-deoxyuridine (BrdU) immunolabeling, we quantified proliferating cells in rats that were subjected to pilocarpine-induced seizures and subsequently treated with BrdU.

At 1 week post-seizure, both the vehicle- and imipramine-treated groups exhibited a significant increase in BrdU+ cells in the subgranular zone (SGZ) compared to sham vehicle-treated rats. However, the number of BrdU+ cells was comparable between the imipramine- and vehicle-treated groups (Figure 5A,B: sham vehicle: 53 ± 7.9, sham imipramine: 47 ± 3.6, seizure vehicle: 118 ± 15.5, seizure imipramine: 133 ± 31.5). This suggests that 1 week of imipramine treatment post-seizure does not significantly affect progenitor cell proliferation.

Further examination at 4 weeks revealed a general decline in BrdU+ cell numbers post-seizure, consistent with previous findings that show an initial increase in proliferation after seizures, followed by a decrease [15,51]. Notably, after 4 weeks, the imipramine-treated group displayed a higher survival rate of the initially labeled BrdU+ cells compared to the vehicle-treated group. This suggests that continuous 4-week imipramine treatment enhances the survival of proliferating cells post-seizure (Figure 5C,D: sham vehicle: 28 ± 2.0, sham imipramine: 25 ± 3.7, seizure vehicle: 26 ± 7.0, seizure imipramine: 62 ± 9.9. Survival ratios: sham vehicle: 53%, sham imipramine: 54%, seizure vehicle: 22.2%, seizure imipramine: 46.3%).

These results indicate that while short-term imipramine treatment does not affect progenitor cell generation post-seizure, prolonged 4-week treatment significantly improves the survival of these cells.

### 3.6. Impact of Imipramine on Spatial Cognition and Memory Post-Seizure

To assess the impact of imipramine on cognitive function and the recovery of spatial cognition following seizures, we utilized the Barnes maze test. The test was conducted over five days, four weeks after the seizure event.

Our results indicated no significant differences in performance between the seizure groups until day 4 of the Barnes maze test. However, from day 5 onward, the seizure imipramine group showed a marked improvement in locating the escape hole compared to the seizure vehicle group, suggesting enhanced spatial cognition (Figure 6A). The experiment controlled for potential physical impairments that could influence movement, ensuring that the observed results reflected true cognitive function.

In the sham groups, no significant differences were found between the sham vehicle and sham imipramine groups, indicating that imipramine did not affect cognitive function in the absence of seizures. However, in the context of seizure-induced cognitive impairment, the seizure imipramine group spent significantly more time in the target quadrant compared to the seizure vehicle group, indicating improved memory recovery (*p* < 0.05, Figure 6B).

To determine whether imipramine’s anxiolytic properties confounded its effects on cognitive function, we analyzed Barnes maze test data to assess locomotor activity across experimental groups. This analysis provided an indirect measure of general activity levels, ensuring that the improvements observed in the seizure imipramine group were not solely due to increased locomotion. Our findings showed no significant differences in overall locomotor activity between the seizure imipramine and seizure vehicle groups, indicating that the observed cognitive improvements were not merely a result of heightened exploratory behavior but rather a direct effect of imipramine on spatial cognition recovery. The recorded times (seconds, mean ± SEM) for each group were as follows: sham vehicle = 55 ± 0.19, sham imipramine = 49.5 ± 1.51, seizure vehicle = 32 ± 2.03, and seizure imipramine = 49.5 ± 1.72 (*p* = 0.11).

Additionally, the software used for data analysis was IBM SPSS Statistics version 29.0.2.0, which ensured reproducibility and accuracy in statistical assessments.

## 4. Discussion

This study highlights the neuroprotective potential of imipramine in a pilocarpine-induced seizure model, demonstrating its ability to mitigate neuronal damage, oxidative stress, apoptosis, and cognitive impairments while enhancing neurogenesis. The findings provide a comprehensive understanding of imipramine’s multifaceted role in addressing the consequences of seizure-induced neuronal injury.

One of the most notable findings is the dependence of neuroprotection on treatment duration. Short-term imipramine treatment (one week) did not significantly improve neuronal survival, as shown by similar NeuN-positive cell counts in the imipramine-treated and vehicle-treated groups. However, extending treatment to four weeks led to a substantial increase in neuronal survival, suggesting that sustained intervention is required for imipramine to exert its full neuroprotective effects. This supports the hypothesis that prolonged modulation of cellular pathways is critical for long-term recovery after neuronal injury.

Imipramine’s impact on oxidative stress was also time-dependent. While no significant reduction in oxidative stress markers was observed after one week, four weeks of treatment significantly reduced 4-hydroxy-2-nonenal (4HNE) levels, a marker of lipid peroxidation. This delayed effect indicates that imipramine gradually modulates oxidative stress responses, likely through its influence on ASMase and ceramide pathways. By attenuating these pathways, imipramine reduces oxidative damage and its downstream effects, offering a promising strategy for mitigating seizure-induced neuronal death.

The role of ASMase and ceramide metabolism in neuronal injury is increasingly recognized, particularly in epilepsy models. Elevated ASMase activity and ceramide levels are strongly associated with apoptosis and neuronal degeneration. Imipramine’s ability to reduce ASMase activity and ceramide levels in the hippocampus suggests that it effectively modulates lipid metabolism to prevent cell death [34,52]. These findings align with broader research, suggesting that targeting lipid dysregulation may be a viable therapeutic strategy for epilepsy and other neurodegenerative conditions [34,43,53]. These findings and discussion need to be expanded in future neurological studies with imipramine, comparing acute and chronic time points. Furthermore, additional studies of genetically mutant ASMase are warranted.

Imipramine also effectively mitigated apoptosis, as evidenced by a significant reduction in the Bax/Bcl-2 ratio. Bax, a pro-apoptotic protein, disrupts mitochondrial integrity and promotes cell death, while Bcl-2 stabilizes cellular homeostasis [44,46]. The reduced Bax/Bcl-2 ratio indicates that imipramine suppresses pro-apoptotic pathways. Additionally, increased phosphorylated Akt (p-Akt) levels, a key regulator of cell survival, suggest that imipramine enhances survival signaling, promoting neuronal recovery and resilience.

A critical aspect of this study is imipramine’s enhancement of neurogenesis. Short-term treatment had minimal impact on progenitor cell proliferation, but four weeks of treatment significantly improved the survival and maturation of newly generated neurons in the hippocampus. Neurogenesis is essential for maintaining brain plasticity and functional recovery after injury. While seizure-induced neurogenesis often leads to maladaptive integration and heightened excitability, imipramine appears to counteract these effects, promoting the functional quality and survival of new neurons. This effect may be mediated by imipramine’s modulation of serotonin and norepinephrine systems and its ability to upregulate neurotrophic factors like brain-derived neurotrophic factor (BDNF).

The interplay among ASMase inhibition [34,43,54], serotonin modulation, and neurogenesis raises important questions. To what extent are imipramine’s effects attributable to each mechanism? For example, the observed reductions in the Bax/Bcl-2 ratio and oxidative stress markers could result from ASMase inhibition, serotonin-mediated neuroprotection, or improved neurogenesis. Future studies should aim to disentangle these overlapping mechanisms using selective inhibitors or genetic models to isolate the contributions of ASMase inhibition, serotonin signaling, and neurogenesis.

Cognitive recovery is another area in which imipramine has demonstrated promise. Our previous study found a significant increase in scores on the Modified Neurological Severity Score after seizures, demonstrating a significant loss of neurological function due to neuronal death following epilepsy. In the Barnes maze test, imipramine-treated animals outperformed vehicle-treated animals in spatial learning and memory tasks. This improvement highlights imipramine’s potential not only to protect against structural damage but also to support functional recovery. The relative contributions of ASMase inhibition, serotonin modulation, and neurogenesis to this recovery remain unclear. However, there is some basic and clinical research on the anxiolytic and cognitive-enhancing effects of citicoline, similar to those of imipramine [55,56,57]. Thus, additional behavioral tests, such as open field tests, should be conducted to address the limitations of these studies. 

Despite its promise, this study has some limitations that warrant consideration. First, the effects of imipramine were monitored over a four-week period, leaving the long-term sustainability of its benefits unclear. Future studies should extend the observation period to evaluate whether its neuroprotective effects persist over time. Second, comparative analyses with other neuroprotective agents could provide valuable insights into imipramine’s relative efficacy and broader clinical applicability. Such comparisons would help clarify its unique contributions and potential advantages over alternative treatments. Although imipramine is widely used as a pharmacological inhibitor of ASMase, it also affects serotonin signaling. To address this limitation, we carefully designed our experiments to focus on endpoints directly related to ASMase activity. Furthermore, we interpreted our results within the context of imipramine’s broader pharmacological profile to ensure accurate conclusions. To further elucidate the physiological role of ASMase and strengthen the mechanistic understanding, future studies should employ selective ASMase inhibitors or genetic knockout models. Additionally, further research should explore the underlying molecular mechanisms by which imipramine modulates lipid metabolism and neurogenesis. Determining the optimal dosage and timing for treatment will also be essential to maximize its therapeutic potential.

## 5. Conclusions

In conclusion, imipramine exhibits significant therapeutic potential in the context of seizure-induced neuronal injury. Its ability to reduce oxidative stress, modulate lipid metabolism, suppress apoptosis, promote neurogenesis, and improve cognitive function positions it as a comprehensive intervention for managing post-seizure complications. By addressing this study’s limitations and pursuing future research, imipramine could pave the way for novel therapeutic strategies in epilepsy and other neurological disorders characterized by neuronal damage.

## Figures and Tables

**Figure 1 cells-14-00281-f001:**
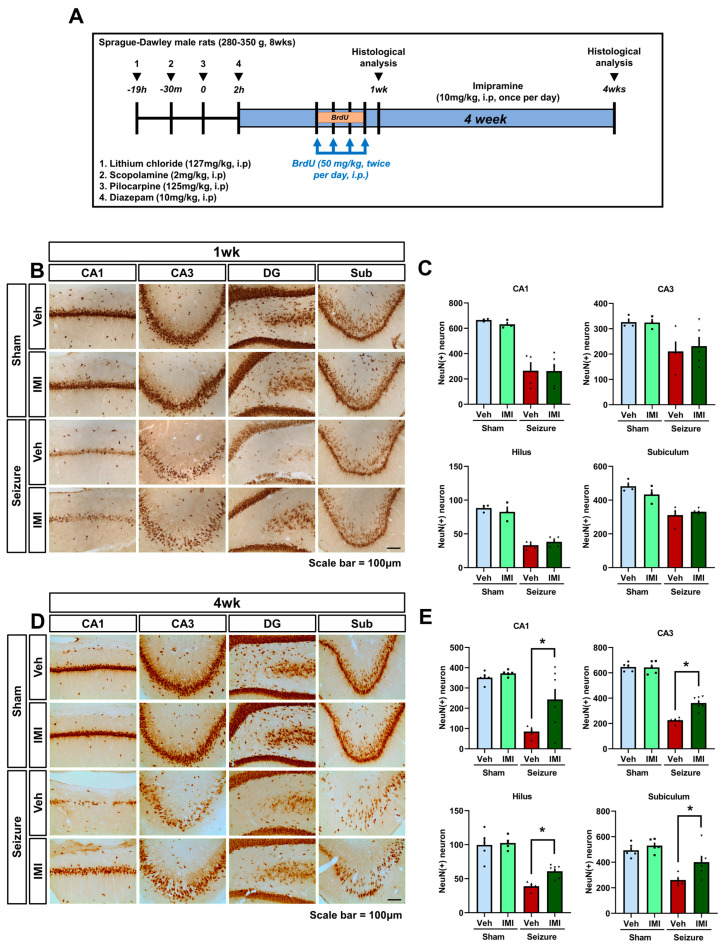
Impact of imipramine on hippocampal neuron survival after seizure. (**A**) Experimental timeline: seizures were assessed for 2 h after pilocarpine administration and when stage 4 (rearing with forelimb clonus) was reached. We also describe the experimental protocol and overall experimental schedule for the different groups (sham vehicle, sham imipramine, seizure vehicle, seizure imipramine) after the seizure episode. (**B**) NeuN staining of hippocampal neurons 1 week after seizure shows substantial cell loss in the CA1, CA3, hilus, and subiculum regions following pilocarpine-induced seizures. Comparison with the 1-week imipramine treatment group shows no significant neuroprotective effect in these regions. Scale bar: 100 μm. (**C**) Quantitative analysis of NeuN-positive neurons in hippocampal regions (CA1, CA3, hilus, and subiculum) shows no significant difference in neuronal mortality between vehicle-treated and imipramine-treated groups at 1 week post-seizure (After 1 week, NeuN: CA1—*p* = 0.9, CA3—*p* = 0.71, hilus—*p* = 0.28, subiculum—*p* = 0.54). (**D**) After 4 weeks, NeuN staining indicates significant neuronal loss in the CA1, CA3, hilus, and subiculum in seizure-induced groups, but 4-week imipramine treatment demonstrates neuroprotective effects in these regions, reducing neuronal loss compared to the vehicle group. Scale bar: 100 μm. (**E**) Bar graph representing the number of NeuN-positive neurons in the CA1, CA3, hilus, and subiculum regions of the hippocampus. Data (n = 5 per group) are presented as mean ± SEM (after 4 weeks NeuN: CA1—*p* = 0.021, CA3—*p* < 0.001, hilus—*p* = 0.001, subiculum—*p* = 0.033). * Significant difference from vehicle group, *p* < 0.05.

**Figure 2 cells-14-00281-f002:**
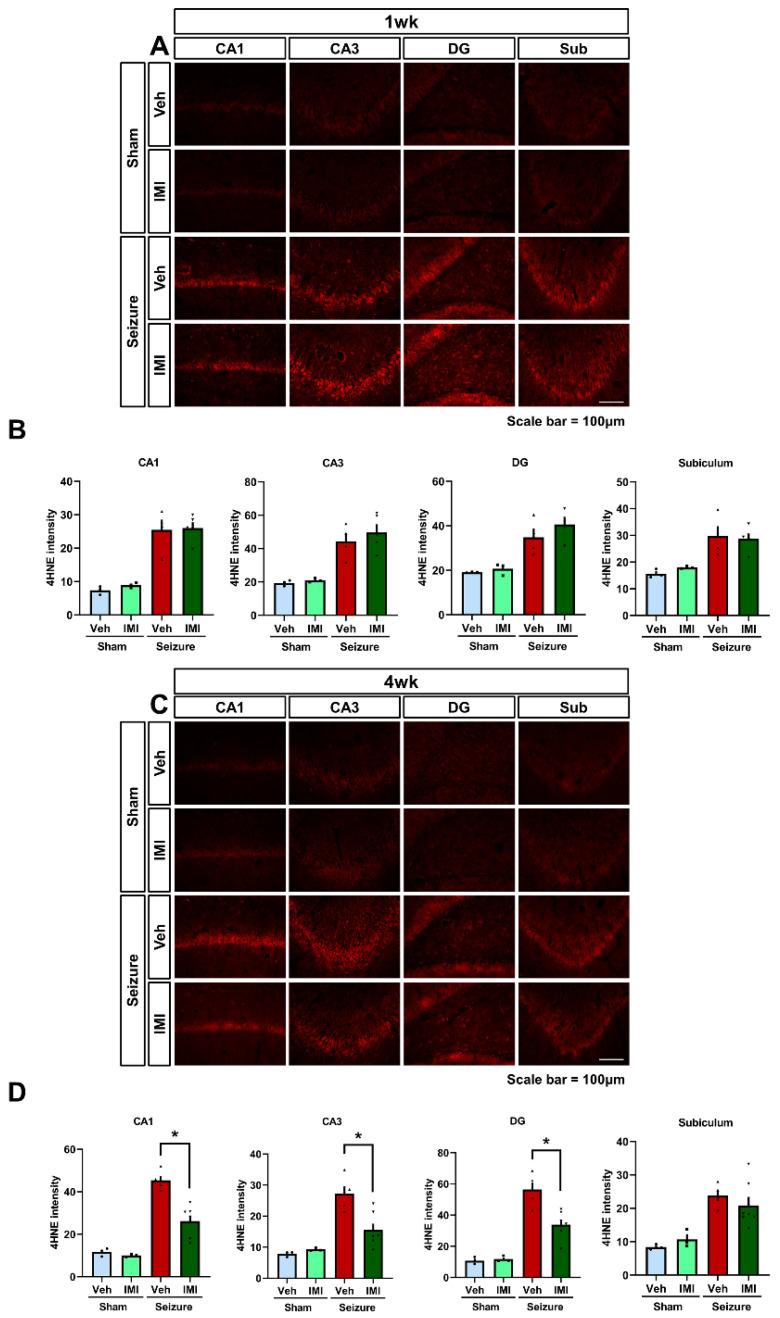
Imipramine decreased oxidative stress after pilocarpine-induced seizure. (**A**) After seizures, 4HNE intensity was elevated in the seizure groups compared to the sham group. The seizure imipramine group displayed 4HNE intensity similar to that of the vehicle-treated group. Scale bar = 100 μm. (**B**) The corresponding bar graph quantifies the 4HNE fluorescence intensity in the hippocampus shown in (**A**) (after 1 week, 4HNE: CA1—*p* = 0.89, CA3—*p* = 0.28, hilus—*p* = 0.47, subiculum—*p* = 0.82). (**C**) Microscopic images demonstrate the extent of oxidative damage through 4HNE staining in various hippocampal regions. A significant increase in 4HNE intensity is observed in the seizure group relative to the sham vehicle group, while the seizure imipramine group shows a reduction in 4HNE intensity compared to the seizure vehicle group. Scale bar = 100 μm. (**D**) The bar graph presents quantitative analysis of 4HNE levels across different hippocampal regions. Data for each group (n = 5) are presented as mean ± SEM (after 4 weeks 4HNE: CA1—*p* = 0.004, CA3—*p* < 0.001, hilus—*p* = 0.003, subiculum—*p* = 0.336). * Significant difference from vehicle group, *p* < 0.05.

**Figure 3 cells-14-00281-f003:**
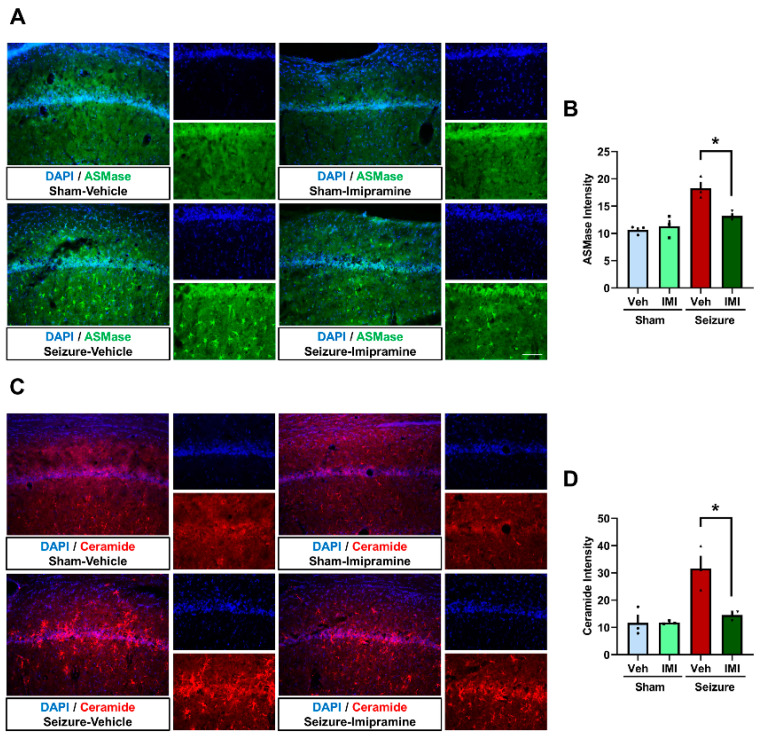
Effect of imipramine on acid sphingomyelinase activity and ceramide production post-seizure. (**A**,**C**) Fluorescent microscopy images displaying the effects of imipramine treatment on ASMase (green) and ceramide (red) activities in the hippocampal CA1 region. The images compare these activities across sham, seizure vehicle, and seizure imipramine treated groups. (**B**,**D**) Bar graphs quantifying the fluorescence intensities of ASMase and ceramide in the CA1 region of the hippocampus. Data are presented as mean ± SEM, with sample sizes of n = 3 for each sham group and n = 3 for each seizure group. A significance level of * *p* < 0.05 was used for statistical analysis (ASMase: *p* = 0.033, ceramide: *p* = 0.061). * Significant difference from vehicle group, *p* < 0.05.

**Figure 4 cells-14-00281-f004:**
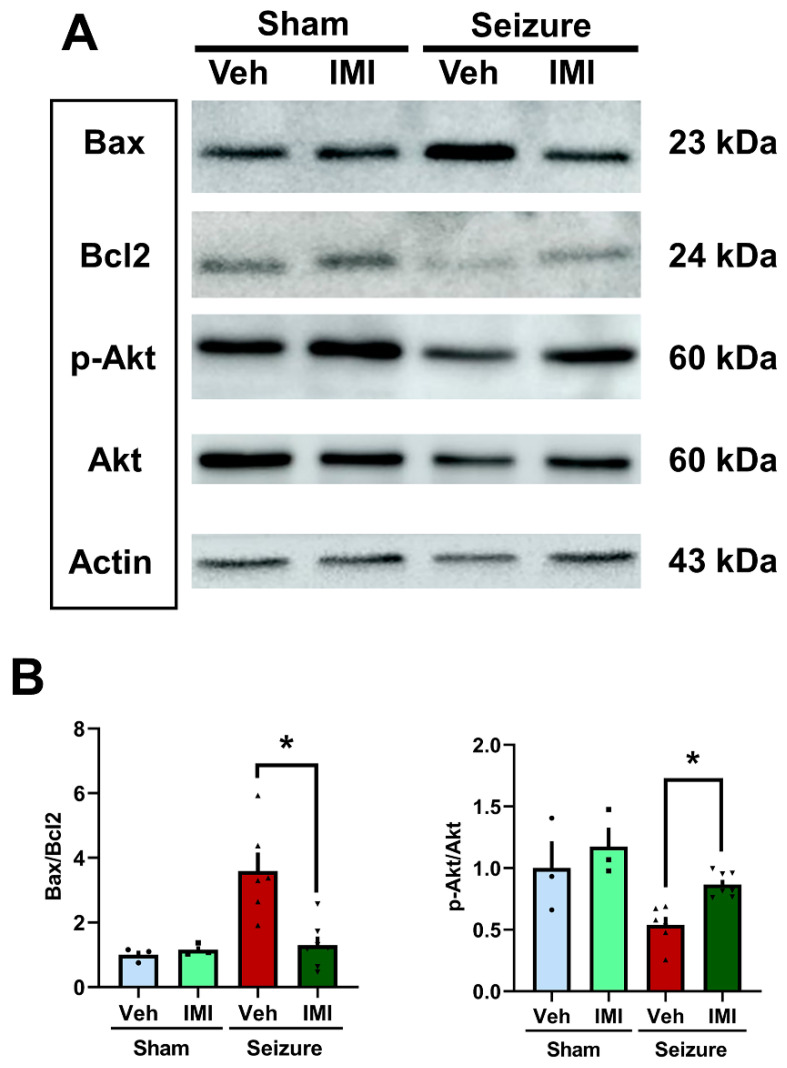
Effects of imipramine on apoptosis and proliferation markers post-seizure. (**A**) Western blot analysis showing the levels of Bax, Bcl-2, p-Akt, and Akt proteins in the hippocampus. In the sham groups, levels of active Bax, Bcl-2, and p-Akt were appropriately maintained. Post-seizure, there was a significant increase in Bax and a decrease in Bcl-2 levels, along with a reduction in p-Akt. Imipramine treatment restored Bax, Bcl-2, and p-Akt levels to near-normal levels. (**B**) The accompanying bar graph quantifies Bax, Bcl-2, and p-Akt expression in the hippocampus. Notable differences were observed between the vehicle- and imipramine-treated groups. Data are presented as mean ± SEM, with n = 5–7 per group. A significance threshold of * *p* < 0.05 was used (Bax/Bcl2: *p* = 0.045, p-Akt: *p* = 0.002). * Significant difference from vehicle group, *p* < 0.05.

**Figure 5 cells-14-00281-f005:**
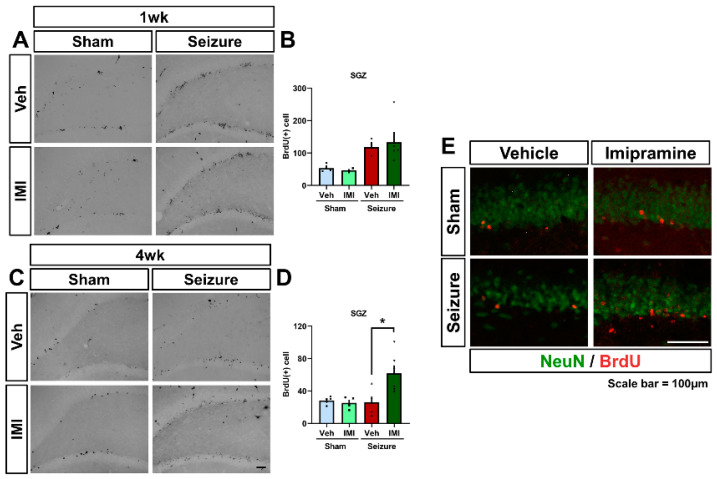
Influence of imipramine on progenitor cell survival post-seizure. (**A**) BrdU immunohistochemistry staining in the subgranular zone (SGZ) of the dentate gyrus (DG) shows neuroblast production. Post-seizure, proliferative BrdU-labeled cells are prominently observed in the SGZ. Both imipramine- and vehicle-treated groups showed increased BrdU+ cells 1 week post-seizure, with detailed views provided in the enlarged boxed regions. Scale bar = 100 μm. (**B**) Bar graph displaying the count of BrdU+ cells in the SGZ, highlighting an increase in the seizure group compared to the sham group, with no significant differences between the imipramine- and vehicle-treated groups (after 1 week, BrdU: *p* = 0.68). (**C**) Depiction of BrdU+ cells in the SGZ of the DG in adult rats 4 weeks post-BrdU injection. Post-seizure, a decline in BrdU-labeled cells is observed. The imipramine-treated group shows a relative decrease in BrdU-labeled cells compared to the seizure vehicle group but retains higher levels than the vehicle-treated group. Enlarged boxed regions are shown, scale bar = 100 μm. (**D**) Quantification of BrdU-labeled cells in the hippocampus, demonstrating a higher count in imipramine-treated rats compared to vehicle-treated rats (after 4 weeks, BrdU: *p* = 0.017). (**E**) Presence of BrdU+NeuN+ cells in the DG region 4 weeks post-BrdU injection, localized in the granule cell layer. Scale bar = 100 μm. Data are presented as Mean ± SEM, with n = 5–8 per group, * Significant difference from vehicle group, *p* < 0.05.

**Figure 6 cells-14-00281-f006:**
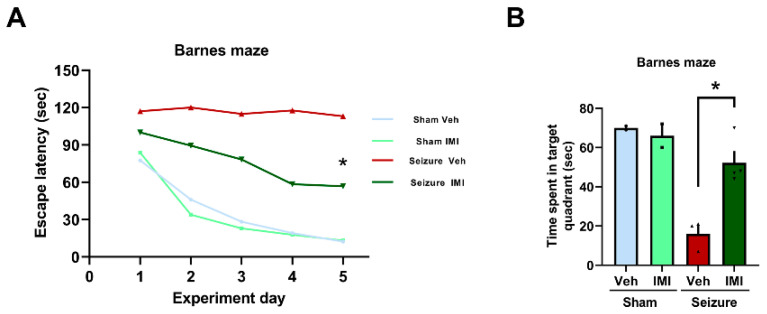
Effect of imipramine on spatial cognition and memory following seizure. (**A**) In the Barnes maze test, seizure-induced groups exhibited reduced spatial cognitive performance compared to sham groups. Notably, the seizure imipramine group demonstrated significantly better performance than the seizure vehicle group, particularly from day 5 onward. No significant differences were observed between the sham vehicle and sham imipramine groups on any testing day (Barnes maze: *p* = 0.008). (**B**) Quantitative analysis of the time spent in the target quadrant indicates improved spatial recognition in the imipramine-treated group compared to the seizure vehicle group. Repeated measures analysis revealed significant group interaction effects (mean ± SEM; n = 5), with a notable improvement in Barnes maze performance (time spent in target quadrant: *p* = 0.004) for the time × group interaction. * Significant difference from vehicle group, *p* < 0.05.

## Data Availability

Data are contained within the article.

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
