# Peer review of "Imipramine, an Acid Sphingomyelinase Inhibitor, Promotes Newborn Neuron Survival in the Hippocampus After Seizure"

_cells, 2025, doi:10.3390/cells14040281_

Round 1
Reviewer 1 Report
Comments and Suggestions for Authors
Acid sphingomyelinase (ASMase), an enzyme that hydrolyzes sphingomyelin into ceramides, is implicated in oxidative stress, neuroinflammation, and neuronal apoptosis. This study aims to determine the role of ASMase in experimental epilepsy, a chronic neurological disorder, that can be triggered by various insults, including traumatic brain injury, stroke, and genetic mutations. The authors used classical pilocarpine model in rats, to test whether seizure activity upregulates ASMase and ceramide levels. They found that pilocarpine increased both ASMase and ceramide levels in the hippocampus, 4 weeks post-seizure induction. They also found that the antidepressant and ASMase inhibitor prevent these seizure-induced increases. The authors also tested the effects of imipramine on neural number, neurogenesis and spatial memory in Barnes test. The authors concluded that "ASMase inhibition could mitigate neuronal apoptosis and improve cognitive recovery after seizures," and proposed that imipramine "may serve as a promising therapeutic strategy for epilepsy-associated neuronal damage and cognitive deficits." While the study’s significance is clear, several key issues must be addressed to strengthen the claims and interpretations.
MAJOR POINTS
1- Imipramine is used here to inhibit ASMase, however, this method is not specific since imipramine also acts as an antidepressant namely by targeting serotonin signaling. While this point is acknowledged and discussed, the authors should validate their findings using another inhibit ASMase inhibitor, especially given the title.
2- As a widely used antidepressant, imipramine likely reduces anxiety. Its effects on cognitive function could therefore be influenced by its anxiolytic properties. The authors should address this possibility and assess anxiety levels across experimental groups using validated behavioral tests, such as the open field test and elevated plus-maze.
3- The authors used the Mann-Whitney U test. They should provide a clear rationale for selecting this non-parametric test in the Methods section.
4- When and how Imipramine was administered should be clearly described. In figure 1 a scheme with a timeline showing seizure induction, Imipramine injections and immunostaining, would greatly improve clarity for readers.
5- The authors tested the effects of imipramine on acid sphingomyelinase (ASMase) activity and ceramide production four weeks after seizure induction. Imipramine beneficial effects on neural survival and oxidative stress are seen at 4 weeks post seizure but not 1 week post seizures (shown in Fig 1 and 2). It is therefore important to compare the effects of imipramine on acid sphingomyelinase (ASMase) activity at 1 vs 4 weeks post-seizure induction. This will help understand the time course of seizure-induced increase in ASMase expression and function (Fig 3).
6- Method section should indicate whether male or female rats were used, and in which proportions.
MINOR POINTS
In Figure 3A and 3B, the data represent ASMase expression rather than its enzymatic activity. This distinction should be explicitly stated in the figure legend and text.
Figure 5 should be resized to match the dimensions of the other figures, ensuring uniform presentation.
All figures should include descriptive titles to improve readability and provide a quick reference for their content.
Is it possible to improve resolution of the fluorescent microscopy images?
Reviewer 2 Report
Comments and Suggestions for Authors
The authors present work to show the neuroprotective effects of imipramine - an ASMase inhibitor in hippocampus in rodent model of seizure. This work argues ASMase as a promising therapeutic target to effectively ameliorate seizure phenotypes. Although some limitations and questions remain, and are included in the comment question below, the presented manuscript is in line within the scope of the journal.
Comments to be addressed:
1. In Abstract, line 15 regarding "genetic mutations": Is there any genetic evidence linking ASMase genetic mutations to vulnerability /resistance to epilepsy?
2. Line 68-70: Please add reference.
3. In Results: Were the pharmacology studies corroborated by any genetic manipulations (KD or KO) to validate it is ASMase-specific neuroprotective effect?
4. In Results: Is the neuroprotective effect mediated by ASM inhibition dose-dependent and time-dependent? Was 10mg/kg imipramine empirically determined as optimal dosage without any off-target effect? Did 4 wk dosing regimen show optimal neuroprotective outcome?
5. In Results: Comparison should also be made between imipramine-treated sham vs seizure animals - to show whether ASMase inhibition alone is sufficient to rescue disease phenotypes (e.g. neuronal loss, oxidative stress, apoptosis, etc.).
6. In Section 3.3: Was ASMase or ceramide IF staining performed at 1wk as well?
7. In Section 3.4: Similarly, was apoptosis or proliferation marker in WB performed at 1wk as well?
8. In Section 3.4: Relative p-Akt level should be quantified as p-Akt/ total Akt ratio.
9. In Section 3.5: Does epilepsy only affect spatial cognitive function? If not, has other short-term or long-term learning & memory test or locomotive activity assessed in the study?
Round 2
Reviewer 1 Report
Comments and Suggestions for Authors
The authors acknowledged the limitations and improved their discussion. However, they are not able to perform the necessary experiment. Therefore, I suggest that the authors use the previously collected data (Barnes test) and quantify locomotor activity across the different groups. This could be an important additional information which can be easily extracted from previously performed experiments. The software used to analyze Barnes test data should be indicated as well.euron Survival in the Hippocampus After Seizure

Round 3
Reviewer 1 Report
Comments and Suggestions for Authors
The authors addressed almost all my concerns. They significantly improve their manuscript. However, the authors should show the data or at least proved values and statistics for the following statement (line 646-650): "Our findings revealed no significant differences in overall locomotor activity between the seizure-imipramine and seizure-vehicle groups, suggesting that the observed cognitive improvements were not merely due to heightened exploratory behavior but rather a direct effect of imipramine on spatial cognition recovery. "
